# PAX Genes in Cardiovascular Development

**DOI:** 10.3390/ijms23147713

**Published:** 2022-07-12

**Authors:** Rebecca E. Steele, Rachel Sanders, Helen M. Phillips, Simon D. Bamforth

**Affiliations:** Bioscience Institute, Faculty of Medical Sciences, Newcastle University, Centre for Life, Newcastle NE1 3BZ, UK; r.e.steele2@newcastle.ac.uk (R.E.S.); rachel.sanders@newcastle.ac.uk (R.S.); helen.phillips@newcastle.ac.uk (H.M.P.)

**Keywords:** *Pax3*, *Pax9*, cardiovascular development

## Abstract

The mammalian heart is a four-chambered organ with systemic and pulmonary circulations to deliver oxygenated blood to the body, and a tightly regulated genetic network exists to shape normal development of the heart and its associated major arteries. A key process during cardiovascular morphogenesis is the septation of the outflow tract which initially forms as a single vessel before separating into the aorta and pulmonary trunk. The outflow tract connects to the aortic arch arteries which are derived from the pharyngeal arch arteries. Congenital heart defects are a major cause of death and morbidity and are frequently associated with a failure to deliver oxygenated blood to the body. The *Pax* transcription factor family is characterised through their highly conserved paired box and DNA binding domains and are crucial in organogenesis, regulating the development of a wide range of cells, organs and tissues including the cardiovascular system. Studies altering the expression of these genes in murine models, notably *Pax3* and *Pax9,* have found a range of cardiovascular patterning abnormalities such as interruption of the aortic arch and common arterial trunk. This suggests that these *Pax* genes play a crucial role in the regulatory networks governing cardiovascular development.

## 1. Cardiovascular Development

The heart is the very first organ to form in development and is responsible for providing the embryo with a sufficient supply of blood, via the aortic arch arteries, to support its growth. Different cells of diverse embryonic origins make up the cardiovascular system. The heart is mostly comprised from mesodermal cells with a contribution from the endoderm for specification and differentiation [1]. Multiple studies have shown that the heart is formed from distinct populations of progenitor cells termed the primary and secondary heart fields (reviewed in [2,3]). Briefly, cells from the first heart field form the left ventricle and are involved in atria formation, whereas cells from the second heart field contribute to formation of the outflow tract, right ventricle and atria. Initially, the outflow tract of the heart is formed with a single lumen, which subsequently becomes divided into the separate aorta and pulmonary trunk following the formation and remodelling of the outflow tract cushions; the precursors to the unidirectional valves that prevent blood flowing back to the heart. A protrusion from the dorsal wall of the aortic sac also contributes to the septation of the outflow tract, along with neural crest cells (NCC) which migrate to these tissues [4]. The aorta is transferred to the left ventricle and the pulmonary trunk links with the right ventricle.

The aortic arch arteries, which comprise the aorta, subclavian and carotid arteries, are derived from the pharyngeal arch arteries which form within the pharyngeal arches. The pharyngeal arches are transient, evolutionary conserved structures in embryonic development and resemble bilateral pairs of swellings surrounding the embryonic foregut, developing in a rostral to caudal sequence [5] (Figure 1A). They are composed of diverse embryonic cell types, with an outer layer of ectoderm, an internal layer of endoderm, and a mesenchymal core composed of both NCC and mesoderm. Each arch is demarcated by endodermal pharyngeal pouches and ectodermal pharyngeal clefts (Figure 1B,C). The pharyngeal arches give rise to disparate specialist structures including vasculature, bone, cartilage, muscle and nerves, and derivatives of the third and fourth pharyngeal pouches: the thymus, thyroid and parathyroid glands, and the ultimobranchial bodies [5].

In mammals there are five pairs of pharyngeal arch arteries, numbered one to four, plus the ultimate artery of the pulmonary arch [6,7]. These arteries form sequentially within the mesenchyme of each pharyngeal arch and link the aortic sac to the dorsal aorta but are not all present at the same time in development. Together, they represent the foundations of the mature circulatory system, and undergo complex remodelling to produce the final asymmetric structure of the aortic arch arteries (Figure 2). This process starts with the first and second arch arteries forming first, at embryonic day (E) 8.5 in the mouse, but become interrupted shortly after with their distal parts subsequently forming the mandibular and hyoid arteries, and the proximal parts becoming the base of the external carotid arteries [7,8]. The third arch arteries form next (Figure 2A), and with extensive growth of the embryo in the anterior-posterior axis, remodel into the common carotid arteries and proximal parts of the internal carotid arteries. The distal parts of the internal carotids are formed from the paired dorsal aorta anterior to the carotid duct, which is the segment of the dorsal aorta between the third and fourth arch arteries and this involutes during the remodelling process. The fourth arch arteries on the right and left develop into distinct arch arteries. On the right, the fourth arch artery becomes the proximal portion of the right subclavian artery in conjunction with the right seventh intersegmental artery. The left fourth arch artery develops into the aortic arch. The ultimate arch artery is the last to form but the first to remodel (Figure 2B). On the right, following rotation and septation of the outflow tract, the vessel thins and disappears. On the left, the ultimate artery becomes the arterial duct, entering the descending aorta level with the left subclavian artery which is formed from the left seventh intersegmental artery (Figure 2C,D). This connection, present only in the foetus, allows blood to bypass the non-functioning, fluid filled lungs and closes shortly after birth, to allow for the delivery of deoxygenated blood to the lungs.

NCC are a multipotent population of neuroectoderm derived cells that delaminate from the dorsal portion of the neural tube, undergo an epithelial-to-mesenchymal transition and migrate to target destinations where they differentiate into key cell types such as melanocytes, smooth muscle cells, craniofacial cartilage and bone and the peripheral nervous system. A sub-population of NCC, the cardiac neural crest cells (cNCC), migrate ventrally through the third, fourth and ultimate pharyngeal arches and contribute to outflow tract septation, formation of the valves and differentiate into the smooth muscle cells that invest around the aortic arch arteries that emanate from the heart [10]. In order to migrate, cNCC must navigate complex cellular and molecular interactions to reach their destination. The extracellular matrix regulates the movement of cNCC through the embryo along an equilibrium of permissive substrate components such as fibronectin and laminin, and non-permissive components such as proteoglycans [11].

## 2. Congenital Heart Defects

Congenital heart defects are a major driver of morbidity in childhood, with global incidence rates of approximately 0.8–1.2% in live-born infants [12,13], affecting the heart itself, the outflow tract region, or the aortic arch arteries. They are usually haemodynamically compatible with embryonic and foetal development as oxygenation of the blood by the lungs is not required. It is only following birth that abnormalities of the heart and the aortic arch arteries prevent oxygenated blood being delivered to the whole body. For example, when the outflow tract fails to separate into the aorta and pulmonary trunk, a single lumen tract remains, known as a common arterial trunk (also known as persistent truncus arteriosus) [14]. Another outflow tract defect, is where the aorta is not transferred to the left ventricle and remains in communication with the right ventricle along with the pulmonary trunk, giving double outlet right ventricle [15]. If the fourth pharyngeal arch arteries do not form correctly this can result in interruption of the aortic arch and an aberrant, frequently retro-oesophageal, right subclavian artery [7]. The importance of cNCC in cardiovascular development was illustrated by the surgical ablation of the neural crest prior to its migration that resulted in conotruncal abnormalities such as common arterial trunk and aortic arch artery patterning defects [16,17]. Furthermore, genetic ablation of NCC using diphtheria toxin or thymidine kinase sensitivity also results in cardiovascular outflow tract and aortic arch artery abnormalities [18,19,20].

## 3. *Pax* Genes in Development

All members of the *Pax* gene family contribute to organ development during embryogenesis through a wide range of actions influencing cell fate, apoptosis and differentiation [21,22]. The *Pax* genes are regulators of gene expression and are characterised by a highly conserved DNA binding paired domain and a C-terminal transcriptional regulatory domain [23]. The *Pax* genes are further subdivided depending on the presence or absence of additional motifs such as an octapeptide region and a paired-type homeodomain. Pax proteins are essential for eye, kidney, thyroid, skeleton, muscle, lymphocyte and pancreas endocrine cell formation, but only *Pax3* and *Pax9* have been found to be crucial in cardiovascular development.

### 3.1. Pax3 in Cardiovascular Development

The genetic alteration of *Pax3* in mouse models causes defects associated with neural crest derivatives and cardiovascular malformations. Pax3 is a 479 amino acid protein which maps to chromosome 1 in the mouse and has 97.5% homology at the protein level to human *PAX3*, which is found on chromosome 2. The Pax3 protein contains a paired-domain, octapeptide motif, and a paired-type homeodomain (Table 1). *Pax3* mRNA transcripts are first identified from E6.5 in mouse embryo development and localised to the dorsal section of the neuroepithelium and part of the dermomyotome by E8.5. *Pax3* becomes more widely expressed throughout other parts of the embryo, such as the NCC, between E9.5 and E12.5 and *Pax3* transcripts are no longer expressed by E17.5 [24,25].

#### 3.1.1. Splotch Mouse Models

*Pax3* is the gene affected in mice where the *Splotch* locus has been mutated. Mice homozygous for *Splotch* mutations display a range of defects affecting the neural tube, NCC and their derivatives, making *Splotch* mutant mice a widely used model to study defects of neural tube development, such as spina bifida, exencephaly, and Waardenburg syndrome [26]. The most prominent cardiovascular phenotype seen across the range of *Splotch* mutant embryos is the failure of outflow tract septation resulting in common arterial trunk [4]. It is the role of *Pax3* within the NCCs that is critical for outflow tract septation, as well as other NCC dependent organs derived from the pharyngeal arches. A key characteristic shared by all variants of *Splotch* heterozygous mutant mice is the presence of a white spot on the belly, caused by improper *Pax3* regulation of NCC-derived melanocyte development [27].

Multiple mutations of the *Splotch* locus, encompassing alterations or deletions that include *Pax3*, have been identified and are known as *Splotch* (*Sp, Sp1H, Sp2H, Sp4H*), *Splotch-retarded* (*Spr*) and *Splotch-delayed* (*Spd*) (Table 2) [28,29,30,31,32], with many other *Splotch* and “*Splotch-like*” mutations also having been studied [29,33,34,35,36,37]. The *Sp* and *Spd* mutations arose spontaneously, and the others were induced through x-ray mutagenesis although *Sp1H* and *Sp2H* likely share the same mutation as both were isolated from the same mutagenised male [26,31]. Mice homozygous for the *Sp* mutation, and also the *Sp1H* and *Sp2H* mutations, die at day 14 of gestation with defects affecting the lumbosacral and cranial neural tube, neural crest-derived tissues and the cardiovascular system [38,39,40,41]. Mice frequently present with a common arterial trunk emanating from the right ventricle and accompanied by an interventricular communication (or ventricular septal defect). The cardiovascular phenotype, however, may vary depending on the mutation within the *Splotch* locus, or the genetic background of the mice analysed. For example, on a mixed C3H/101 × C57Bl6 background, the majority of *Sp1H* homozygous mutants have common arterial trunk, although occasionally some embryos may only present with double outlet right ventricle [42]. Mutant *Sp1H* embryos also display aortic arch artery patterning defects such as abnormal regression of the ultimate arch artery, retro-oesophageal right subclavian artery and aberrant formation of the common carotid arteries, although these are seen with variable penetrance [42]. Defects affecting the derivatives of the third and fourth pharyngeal arches are also observed in *Sp1H* mutants, including aberrant positioning or non-formation of the thymus, absent ultimobranchial bodies, and malformations affecting the thyroid and parathyroid glands [42]. The *Sp2H* allele also causes variable penetrance of cardiovascular defects in homozygous mutants. Although all *Sp2H/Sp2H* mutants, on a mixed C3H/101 × CBA/Ca background, present with neural tube defects, either spina bifida, exencephaly, or both, only 60% of mutants have an outflow tract defect, with the majority having a common arterial trunk, and a minority with double outlet right ventricle, all associated with an interventricular communication. The remaining 40% of *Sp2H/Sp2H* mutants do not have cardiovascular defects but die perinatally with signs of cardiac failure [43]. The caudal pharyngeal arch arteries are also hypoplastic [44].

Mutant mice with a large chromosomal deletion that includes the *Splotch* locus die early in development, before the cardiovascular system has formed. Mice homozygous for the *Sp4H* mutation have arrested development around E6.0 and an increased proportion of embryo resorptions are seen [29,46]. This early lethality of *Sp4H* mutants is believed to be influenced by the deletion of another gene within the affected region rather than solely due to the loss of *Pax3*, since the null allele found in *Sp1H* and *Sp2H* mutant mice does not cause early embryonic death [46]. For example, mice deficient for *Cul3*, which is in the deleted region, die before E7.5 with defects in mitosis [53]. *Spr* homozygotes are lethal before the preimplantation stage of development [45] possibly due to the deletion of surrounding flanking markers such as *Vil1* and *Inha*, which are essential for the progression to implantation [45,47].

Of all the *Splotch* mutations, the *Spd* phenotype can be described as the least severe, with homozygous mutants typically surviving to full term before dying perinatally with a reduction in the size and number of spinal ganglia [54]. Spina bifida also develops due to the abnormality in interactions between the layers of embryonic tissues leading to faults with neural tube closure. *Spd* homozygous mutants also typically present with double outlet right ventricle, and in *Sp1H/Spd* double heterozygous compound mutants around 50% of fetuses had common arterial trunk [54].

#### 3.1.2. *Pax3* Mouse Models

An engineered genetic alteration of the *Pax3* allele that deletes exon 5 (*Pax3^Δ5^*), creating a premature stop codon and loss of the Pax3 homeodomain, is genetically analogous to the *Sp2H* allele [55]. Homozygous mutants, on a C57BL6 background, display a cardiovascular phenotype of common arterial trunk and interventricular communication with 100% penetrance at E14.5 [20]. In these mutants, malformations of the caudal pharyngeal arch arteries were seen at E11.5, with abnormal regression of the left ultimate arch artery in the mutants which may explain the formation of the common arterial trunk [20]. Additionally, cNCC migration was disorganized with reduced numbers seen in the pharyngeal arches and outflow tract [20].

Through targeting of cDNA sequences of the transcription factor *Forkhead Box O1* (*FKHR*) in the *Pax3* locus of mouse embryonic stem cells, a *Pax3-FKHR* knock-in allele and consequential Pax3-FKHR fusion protein was generated [52]. In this fusion protein, the C-terminal transcriptional transactivation sequence of Pax3 is replaced with the bisected DNA binding domain of FKHR, which is a stronger transactivating sequence, giving the fusion protein the ability to bind homeodomain sites alone. Interestingly, heterozygous offspring of the chimeric mice were found to die at around the time of birth with a range of developmental abnormalities including cardiovascular defects partially resembling those seen in *Sp2H* mice [43]. The fusion protein was expressed at a significantly reduced level compared to the wild-type expression of *Pax3* in embryogenesis, suggesting that a dominant negative effect on the *Pax3* gene is contributing to the observed phenotype. It was deduced that the cause of neonatal death was a combination of defects such as ventricular septal defects and tricuspid valve insufficiency, ultimately causing congestive heart failure.

A novel *Splotch* allele was created by replacing the first exon of *Pax3* with the *Cre recombinase* gene. Homozygous mutants on a mixed C57BL/6 × 129Sv genetic background displayed the typical *Splotch* phenotype although the outflow tract defects of common arterial trunk and double outlet right ventricle observed were not fully penetrant [48]. The *Pax3Cre* allele also allowed for lineage tracing and identified novel *Pax3* expression in hindgut and urogenital epithelium.

#### 3.1.3. *Pax3* Interaction with *Pax7*

Mutant mice with a hypomorphic allele of *Pax3* (*Pax3^neo^*), which produces only 20% of normal transcripts when homozygous, do not have neural tube or cardiovascular defects [49]. *Pax3^neo^;Pax3**^Δ^**^5^* mutants with 10% of normal *Pax3* levels, however, have neural tube defects, but the outflow tract of the heart was septated normally [50]. *Pax3* shares some overlapping embryonic expression domains with *Pax7* in NCC (9,10). Although both these genes share a similar protein structure (Table 1), *Pax7*-null mice are phenotypically unaffected [56]. *Pax3* acts to repress *Pax7* in normal development as in *Pax3* mutants *Pax7* is upregulated [50,56]. Mice double null for *Pax3* and *Pax7* die earlier than *Pax3*-nulls at E11.5 with more severe neural tube defects, suggesting redundancy between these two genes [57]. Compound mutant mice were engineered to produce 10% levels of *Pax3* and reduced amounts of *Pax7* [50]. In *Pax3^neo^;Pax3**^Δ^**^5^**;Pax7^+/−^* embryos common arterial trunk and double outlet right ventricle phenotypes were observed in half of the mutants examined. When embryos were produced that were also deficient for *Pax7* (i.e., *Pax3^neo^;Pax3**^Δ^**^5^**;Pax7^−/−^*), however, all of the mutants presented with common arterial trunk. This study therefore demonstrated that a threshold of *Pax3* expression of >10% is required for correct outflow tract formation, and *Pax7* compensates for loss of *Pax3* in cardiovascular development through upregulated expression [50].

#### 3.1.4. *Pax3* Interaction with *Foxd3*

*Foxd3* is a forkhead transcription factor, and similar to *Pax3*, is an early regulator of neural crest progenitor cells. *Foxd3* deficiency results in early embryonic lethality at around E6.5 [58], and conditional deletion from the NCC leads to a reduction in the number of NCC and cardiovascular defects such as abnormal aortic arch arteries and common arterial trunk in a minority of mutants [59,60]. When *Pax3* expression is heterozygous in conjunction with *Foxd3* loss in the NCC, however, common arterial trunk is seen in all mutants [60] and resembles the phenotype seen in the NCC ablation model [18]. Foxd3 directly binds to enhancers in the *Pax3* locus [61] and *Foxd3* expression is also reduced in *Pax3^Sp/Sp^* mutants [62], suggesting it functions downstream of *Pax3* and therefore may be involved in a feedback loop mechanism.

#### 3.1.5. *Pax3* Interaction with *Msx2*

*Msx2* is a transcription factor expressed in a range of tissues and organs, including the dorsal neural tube and the neural crest [63]. *Msx2*-deficient mice have defects in endochondral bone formation as well as aberrant tooth and mammary gland development [64]. *Msx2* is functionally redundant with *Msx1* (which interacts with *Pax9* as described below), and mice lacking both *Msx2* and *Msx1* have cardiovascular defects [65,66,67]. Whilst *Splotch* mutants die at E13.5, *Pax3;Msx2* double null mutants survive to the neonatal stage and do not present with cardiovascular defects [68]. In normal development, Pax3 binds to the promoter of *Msx2* to repress its expression in a NCC-specific manner, and in the absence of *Pax3*, *Msx2* is upregulated and disrupts NCC development [68].

#### 3.1.6. *Pax3* Interaction with *Sox10*

*Sox10*, an SRY-box transcription factor, physically interacts with *Pax3* [69] and mutations in both genes are known to be involved in Waardenburg syndrome in humans [70]. Patients typically present with NCC related defects such as hearing loss and pigmentation abnormalities, and in rare cases, also congenital heart abnormalities such as atrial and ventricular septal defects [71,72].

#### 3.1.7. *Pax3* Interaction with *Tbx18*

*Tbx18* physically interacts with *Pax3* to cooperatively regulate gene expression in the paraxial mesoderm [73]. *Tbx18* deficient mice die perinatally with multiple defects that include the pleuropericardial membranes, the sinus node, epicardium and coronary vasculature of the heart [74,75,76,77].

#### 3.1.8. *Pax3* and Neural Crest Cells

RNA analysis experiments have demonstrated that the expression of *Pax3* can be used as a marker of cNCC in the mouse embryo until the time of their entry into the outflow tract [43]. In homozygous *Sp2H/Sp2H* embryos the mutant variant of *Pax3* is expressed in the neural tube, but migration of cNCC at E10.5 is defective with reduced cell numbers seen to migrate through the third, fourth and ultimate pharyngeal arches and a failure to arrive at their intended destination in the outflow tract [43]. The role of Pax3 in the migration and function of cNCC seems to be cell autonomous, as homozygous *Sp/Sp* mutant mice, with a transgenic overexpression of *Pax3* in cNCC driven by a 1.6kb *Pax3* regulatory region, allowed for a phenotypic rescue of neural tube closure and cardiac developmental defects [40]. Conditional deletion of *Pax3* from the neural crest, however, does not fully recapitulate the common arterial trunk defect [20]. When *Pax3* is deleted using the *Wnt1Cre* allele, which is active in migrating NCC, no outflow tract defects were seen, although just over half of all mutants displayed exencephaly. When an *AP-2aCre* allele was used, which is active earlier within the neural folds, all mutant embryos developed double outlet right ventricle and died perinatally. *Pax3* is therefore required as a neural border specifier that is essential for premigratory NCC induction [78]. NCC in *Splotch* mutant mice proliferate normally but show altered migratory characteristics indicating that *Pax3* is required for cNCC migration, perhaps through chemotropic responses to migratory cues or through the directionality and persistence of cell movement [79]. This may be controlled through an inhibition of *p53* dependent processes during cNCC migration as it has been shown that neural tube closure defects and an increased rate of apoptosis in *Sp/Sp* embryos is *p53* dependent [80,81]. Interestingly, the inactivation of *p53* in *Pax3*-deficient embryos (i.e., *Pax3^−/−^;p53^−/−^*) leads to a rescue of the cNCC migration and outflow tract septation defects seen in *Pax3*-null embryos [81]. Furthermore, *Pax3* inhibits p53 protein stability through stimulation of ubiquitination and degradation pathways independently of DNA-binding and transcriptional regulation [82]. Taken together, this suggests that *Pax3* acts in the cNCC to inhibit p53-dependent processes that lead to apoptosis, thus allowing for optimal migration of cNCC to enable normal septation of the outflow tract of the heart. The upstream region of the *Pax3* gene in mice contains two evolutionary conserved elements that are vital for *Pax3* expression in the neural crest [83,84]. Mice engineered to replace this neural crest element (NCE), with a loxP flanked neomycin cassette, creates a null or severely hypomorphic *Pax3* allele and homozygous mutants develop neural tube defects and common arterial trunk, thereby recapitulating the phenotype seen in the majority of *Splotch* mutants [51]. When the neomycin gene is removed, however, homozygous *Pax3^NCE^* mutant mice develop normally indicating that the neomycin cassette itself was disrupting *Pax3* expression.

### 3.2. Pax9 in Cardiovascular Development

*Pax9* is a transcription factor important for multiple aspects of embryogenesis, controlling craniofacial, skeletal and pharyngeal development [85,86] and has more recently been shown to be crucial for cardiovascular morphogenesis [87,88,89]. In humans, *PAX9* is found on chromosome 14 and consists of 342 amino acids, with an N-terminal paired DNA-binding domain, an octapeptide motif, but no additional homeodomain (Table 1). In the mouse, *Pax9* is found on chromosome 12 and has 343 amino acids. Human and mouse PAX9 have 98% homology at the protein level.

#### 3.2.1. *Pax9* in the Pharyngeal Endoderm

*Pax9* is expressed throughout the developing embryo beginning at E8.5 in the mouse in the foregut epithelium and becoming specific to the pharyngeal endoderm by E9.5 (Figure 1D). Pax9 is expressed in the endoderm of the first four pharyngeal pouches by mid-embryogenesis, before eventually being expressed throughout the developing embryo in the craniofacial region and skeleton [90]. In equivalently staged human embryos, however, expression was only seen in the third and fourth pouches [91]. *Pax9* is critical for skeletal and craniofacial development as mice deficient for *Pax9* die perinatally with preaxial digit duplications, cleft secondary palate and oligodontia. *Pax9*-null mice also lack the thymus, parathyroids and ultimobranchial bodies, all derivatives of the third and fourth pharyngeal pouches which arrest in development by E11.5 [85]. Mice deficient for *Pax9* on a C57Bl/6 genetic background die neonatally from severe cardiovascular defects, which include double-outlet right ventricle, interrupted aortic arch, retroesophageal origin of the right subclavian artery, and absent common carotid arteries (Figure 3) [88]. They also have hypoplasia of the ascending aorta and bicuspid aortic valve. The phenotypes involving the aortic arch and the subclavian artery are due to bilateral failure of the fourth arch arteries to form (Figure 3E–G). Although no complete vessel is observed, it is possible to find isolated endothelial cells within the fourth arches. The absence of the common carotid arteries reflects the collapse of the third arch arteries (Figure 3E–G). These vessels form bilaterally, but by E11.5, during the phase of remodelling, the vessels begin to degenerate, at least in part, through the lack of investment of the smooth muscle cells derived from the neural crest. This may relate directly to the significant reduction in the number of NCC migrating into the pharyngeal arches. Indeed, in another genetic mouse model where *Msx1* haploinsufficiency was coupled with *Pax9* deficiency (see below), the number of migrating NCC was normalized, the arteries of the third arch were invested with smooth muscle cells, and the common carotid arteries were preserved [89]. Interestingly, on a CD1 genetic background, *Pax9*-null mice had a significantly lower incidence of double outlet right ventricle and bicuspid aortic valve compared to those on the C57Bl/6 background, although the penetrance of arch artery defects was similar [89]. This suggests that genetic modifiers may play a role in the penetrance of the *Pax9* related outflow tract defects, perhaps by influencing second heart field progenitor cells. Inactivation of *Pax9* in NCC using *Wnt1Cre* mice and *Pax9*-floxed alleles caused cleft secondary palate and tooth agenesis and revealed that the *Pax9* expressing mesenchymal cells of the nose, palate, and teeth are derived from NCC [92]. *Pax9*, however, does not have a cell autonomous role within the cNCC, as *Wnt1Cre* conditional deletion of *Pax9* does not result in any cardiovascular defects [88].

In normal development, the arteries of the first and second arches have usually remodelled by E10.5 but in *Pax9*-null mutants these vessels aberrantly persist, being observed either bilaterally or unilaterally (Figure 3E,F) [88]. Whether expression of *Pax9* is directly important for this aspect of remodelling, or if the changes are a consequence of the altered haemodynamics due to absence of the arteries of the fourth arch and collapse of those of the third, is currently unknown. Alteration of blood flow at mid-embryogenesis in both the mouse and chick has been shown to affect the morphogenesis of the arch arteries [93,94]. In the absence of the third and fourth arch arteries in the *Pax9*-null mutants, it is the persisting arteries of the first or second arches that form the external carotid arteries, which then arise directly from the ascending aorta (Figure 3H). The internal carotid arteries take their origin directly from the dorsal aorta, this vessel retaining its continuity due to the failure of involution of the carotid duct.

Cardiovascular developmental defects have been associated with patients harbouring chromosomal abnormalities such as deletions and translocations that include *PAX9* Table 1). Patent foramen ovale, patent arterial duct and pulmonary hypertension have all been reported [95,96,97], alongside craniofacial defects or oligodontia. One particularly interesting case featured a patient with cardiovascular phenotypes similar to the *Pax9*-deficient mouse, with interrupted aortic arch, a bicuspid aortic valve, hypoplastic aorta and a ventricular septal defect [98]. The patient had a small (105kb) hemizygous deletion at 14q13 which removed just three genes: *PAX9*, *NKX2-1* and *NKX2-8*. Mice deficient for either *Nkx2-1* or *Nkx2-9* (which is the *NKX2-8* gene in mice) do not display any cardiovascular abnormalities [99,100]. *PAX9*, therefore, may be considered as a potential candidate gene for human congenital cardiac defects.

#### 3.2.2. *Pax9* Interaction with *Tbx1*

It is unlikely that *Pax9* acts alone in the origination of signalling events from the pharyngeal endoderm. Instead, it is expected to function in a genetic network with co-regulated genes, as has been demonstrated by studies looking at the transcriptome of *Tbx1*-null embryos [101,102] where *Pax9* was identified as being one of the many genes downregulated in this model. Analysis of the *Pax9*-null pharyngeal arch transcriptome revealed a reduction in *Tbx1* expression, as well as a significant overlap with genes differentially expressed in *Tbx1*-null embryos, suggesting that *Tbx1* and *Pax9* might share a genetic network [88].

Hemizygosity of *Tbx1* is understood to underly the cardiovascular defects in 22q11 Deletion Syndrome patients. Also known as DiGeorge Syndrome, this is the most common microdeletion syndrome with an incidence of 1:4000 births, and patients display a wide range of clinical pathologies including congenital cardiovascular defects [103,104]. Mice null for *Tbx1* display cardiovascular developmental defects such as common arterial trunk, caused by the failure of the pharyngeal apparatus to form normally [105,106]. Mice heterozygous for *Tbx1* present with defective fourth arch artery formation which leads to interruption of the aortic arch in a minority of mice, but aberrant right subclavian artery is more frequently seen [7,107]. To test for a genetic interaction between *Pax9* and *Tbx1*, mice heterozygous for each gene were crossed together [88]. Only two *Pax9;Tbx1* double heterozygous mice were identified at weaning (the expected number was 42), with the majority subsequently found to die on the day of birth with a much higher incidence of interruption of the aortic arch than seen in *Tbx1* heterozygous mice. *Tbx1* is expressed in all three germ layers of the pharyngeal arches—the endoderm, mesoderm and ectoderm. To confirm the site of the potential *Tbx1-Pax9* interaction, a novel conditional *Pax9Cre* allele was created in mice and used in conjunction with *Tbx1*-floxed allele mice. Mice with a heterozygous conditional deletion of *Tbx1* in the endoderm, which was simultaneously heterozygous for *Pax9*, displayed cardiovascular defects thereby demonstrating a functional interaction between the two genes in the pharyngeal endoderm [88].

#### 3.2.3. *Pax9* Interaction with *Gbx2*

The transcription factor *Gbx2* is downregulated in *Tbx1*-null embryos [102] and has also been shown to genetically interact with *Tbx1* [108]. *Gbx2* is expressed in the pharyngeal endoderm concomitantly with *Pax9* and *Tbx1,* and *Gbx2* expression is downregulated in the pharyngeal pouch endoderm in *Pax9*-null embryos at E9.5 [87,88]. Given the identified functional relationship between *Pax9* and *Tbx1*, the potential for *Gbx2* to be a shared downstream target in a regulatory network with these genes was explored. A proportion of mice null for *Gbx2* display cardiovascular defects such as aberrant right subclavian artery, double outlet right ventricle and interrupted aortic arch [109] although the latter defect was not observed in our study, but a left-right patterning defect was identified, with these differences attributed to the use of a different *Gbx2* mutant allele [87]. In our study we investigated a genetic interaction between *Pax9* and *Gbx2* and found that fewer *Pax9;Gbx2* double heterozygous mice than expected survived to weaning (59 observed, 77 expected) and had cardiovascular defects including interrupted aortic arch and aberrant right subclavian artery. When *Gbx2*-null mice were combined with *Pax9* heterozygosity (i.e., *Gbx2^−/−^;Pax9^+/−^*) there was a statistically significant increase in the incidence of cardiovascular defects in total, and interrupted aortic arch specifically. Moreover, common arterial trunk was also observed in complex compound mutants (i.e., *Gbx2^−/−^;Pax9^+/−^* and *Gbx2^+/−^;Pax9^−/−^*), a defect not seen in either *Pax9*-null or *Gbx2*-null embryos. Collectively our data demonstrated that *Pax9* and *Gbx2* genetically interact in cardiovascular development, with defects seen in double heterozygous mice, and novel and more severe cardiovascular defects in compound mutants [87].

#### 3.2.4. *Pax9* Interaction with *Msx1*

Pax9 is co-expressed with Msx1 during craniofacial development, and mice null for *Msx1* display overlapping palate and tooth phenotypes with *Pax9*-deficient mice [85,110]. *Msx1* is a highly conserved homeobox gene that plays a role in transcriptional regulation in early development. Unlike *Pax9*-null mice, however, *Msx1*-null mice have a phenotypically normal cardiovascular system [110], despite being expressed in the endocardial cushions at E10.5 [111]. *Msx1* is functionally redundant with *Msx2*, and mice deficient for *Msx1* and *Msx2* (i.e., *Msx1^−/−^*;*Msx2^−/−^*) die in late gestation with outflow tract defects as well as craniofacial malformations [65,66,67]. Msx1 and Pax9 proteins interact in in vitro assays [112], and compound *Pax9* and *Msx1* mutant mice demonstrate a synergistic interaction between these two proteins affecting lip and tooth development [113,114]. Pax9 and Msx1 are not co-expressed in the same cell types in the pharyngeal region at mid-embryogenesis in the mouse, however, as Pax9 is restricted to the endoderm, and Msx1 to NCC [85,89,111,115]. These two genes, however, do appear to influence cardiovascular development, despite the lack of overlapping expression in the pharyngeal arches, with *Msx1* haploinsufficiency modifying the *Pax9*-null cardiovascular phenotype. In *Pax9*-null mice the fourth arch artery fails to form and the third arch arteries collapse, concomitant with a reduction in cNCC migration to the pharyngeal arches [88]. As described above, this leads to aortic arch artery defects such as interrupted aortic arch and retro-oesophageal right subclavian artery, from the absent fourth arch arteries, and absent common carotid arteries, from the loss of the third arch arteries. In mice deficient for *Pax9* and simultaneously heterozygous for *Msx1* (i.e., *Pax9^−/−^;Msx1^+/−^)*, a significant reduction in the incidence of interrupted aortic arch and retro-oesophageal right subclavian artery when compared to *Pax9*-null mice was found [89]. Conversely, there was a significant increase in the less severe arch artery defects of cervical origins of the right subclavian artery and aortic arch. These latter defects arise when the fourth arch arteries are absent, but the third arch arteries are maintained and the carotid ducts persist, and these cervical vessels are theoretically compatible with a functional systemic circulation. These phenotypes were coupled with a rescue of the cNCC migrating into the third pharyngeal arches and therefore able to stabilise the third arch arteries by differentiating into the smooth muscle cells that envelop the remodelling vessels [89]. *Pax9^−/−^;Msx1^+/−^* mutant mice, however, did not survive after birth possibly through breathing defects associated with defective hyoid bone and thyroid cartilage formation [89]. Whilst it can be deduced that *Msx1* and *Pax9* interact in cardiovascular development, the mechanisms underlying this interaction are yet to be uncovered and it is anticipated that there are further genetic interactions involving other key genes involved in this.

#### 3.2.5. *Pax9* Gene Dosage

*Pax9*-heterozygous mice (i.e., *Pax9^+/−^*) are healthy and viable, suggesting that haploinsufficiency of *Pax9* does not affect development. We reasoned that a further reduction in *Pax9* levels may reveal a dosage sensitivity threshold that is crucial for correct cardiovascular development, as has been described with genetically altered *Tbx1* alleles in mice [116]. Reduced levels of *Pax9* were achieved using a hypomorphic allele of *Pax9*, which has a neomycin resistance cassette inserted between the second and third coding exons [117]. Mice homozygous for the hypomorphic allele (*Pax9^neo/neo^*) have ~20% of wild type *Pax9* mRNA transcript levels and develop normally, except for displaying oligodontia. Interestingly mice heterozygous for the null allele have 32% of wild type *Pax9* mRNA transcript levels instead of the predicted 50% levels. This indicates there are as yet undescribed feedback mechanisms at the molecular level that regulate *Pax9* transcription during development. When the hypomorphic allele mice are crossed with *Pax9^+/−^* mice, the resulting *Pax9^−/neo^* mice have less than 10% wild type *Pax9* mRNA levels [117] and, on a C57Bl/6 genetic background, do not survive long after birth.

To investigate these mice in more detail for a cause of death we examined *Pax9^−/neo^* embryos at E15.5 by magnetic resonance imaging (*n* = 14) and by intracardiac ink injection at E10.5 (*n* = 3). At E15.5, 64% of mutants (9/14) had a detectable cardiovascular defect, which all involved an aberrant right subclavian artery, either alone (3/14), with interrupted aortic arch (4/14), a right sided aorta (1/14), or double outlet right ventricle (1/14) (Figure 4B,C). All mutants presented with an abnormal thymus, where the lobes were either split and asymmetrically placed (9/14), hypoplastic (3/14) or absent (2/14). At E10.5, all fourth pharyngeal arch arteries (two per embryo, left and right) of the *Pax9^−/neo^* mutant embryos were abnormal, being either hypoplastic (1/6) or absent (5/6) (Figure 4E,F). These data demonstrated that even with very low levels of *Pax9* mRNA, the cardiovascular system can develop without a high penetrance of abnormalities, although it is not known why all *Pax9^−/neo^* mutants die perinatally when a proportion show no signs of a severe arterial duct-dependent cardiovascular defect. It is, however, possibly due to the same mechanism that causes the death of *Pax9;Msx1* mutants described above, but this requires further investigation. All embryos at E10.5 displayed bilateral fourth arch artery defects, but this incidence was not reflected in the mutants at E15.5, where only 64% had defects derived from the fourth arch arteries. This is analogous to the situation in *Tbx1*-heterozygous mice, where significantly fewer foetuses and neonates have fourth arch artery defects compared to those at mid-gestation suggesting that this vessel has the capacity to recover during development [7].

#### 3.2.6. *Pax1* Does Not Interact with *Pax9* in Cardiovascular Development

*Pax1* has a similar protein structure (Table 1) and high sequence homology to *Pax9*, they share promoter binding sites [90] and have overlapping expression in the pharyngeal endoderm in a range of organisms [86,118,119,120,121]. *Pax1*-null mice have skeletal defects affecting the vertebral column, sternum and scapula [122]. Mice homozygous null for *Pax1* and *Pax9* display developmental defects of the vertebral bodies, intervertebral discs and ribs [86], a more severe phenotype, therefore, than seen in *Pax1*-null mice. Patients suffering from Jarcho-Levin syndrome have severe skeletal defects and have reduced levels of Pax1 and Pax9 protein expression [123]. Using genetically mutated mice we have looked to see if *Pax1* and *Pax9* interact in cardiovascular development. We found that the heart and arch arteries of *Pax1*-null mice are normal, and loss of one or two *Pax1* alleles in the context of *Pax9* deficiency, does not alter the typical *Pax9*-null cardiovascular phenotype (Figure 5). *Pax1* and *Pax9* therefore do not genetically interact in cardiovascular development.

## 4. Concluding Remarks

Despite the *Pax* family consisting of nine genes all involved in embryonic development, only *Pax3* and *Pax9* play major roles in cardiovascular development involving NCC deficiencies, but through expression in different tissues. *Pax3* acts cell autonomously in the NCC and is crucial for their migration and the subsequent septation of the outflow tract. In contrast, *Pax9* is vital for pharyngeal arch artery morphogenesis possibly through influencing NCC migration non-cell autonomously to the pharyngeal arches. *Pax9*, furthermore, is also a major contributor in a complex genetic regulatory pathway underpinning morphogenesis of the aortic arch arteries that originates within the pharyngeal endoderm.

## Figures and Tables

**Figure 1 ijms-23-07713-f001:**
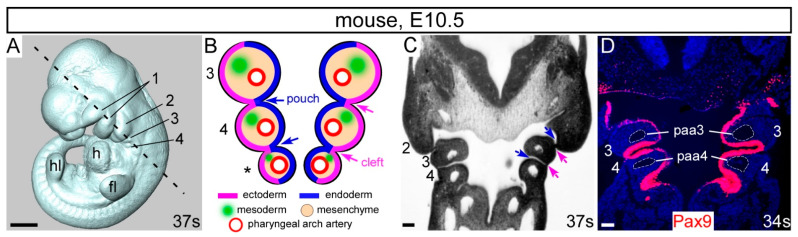
The pharyngeal arches. (**A**) Mouse embryo at E10.5 with 37 somites (s) with the pharyngeal arches 1–4 labelled. (**B**) Schematic coronal overview of the tissues and structures of the 3rd, 4th and ultimate (*) pharyngeal arches. Pharyngeal arches are numbered, and pharyngeal pouches (blue arrows) and pharyngeal clefts (pink arrows) indicated. Each pharyngeal arch is filled with neural crest-derived mesenchyme. (**C**) Coronal section of the embryo in (**A**) with the section plane shown with a dotted line. (**D**) Coronal section of an E10.5 embryo immuno-stained with an anti-Pax9 antibody, specifically labelling the pharyngeal endoderm. The 3rd and 4th pharyngeal arch arteries are indicated. Abbreviations: ec, ectoderm; en, endoderm; fl, forelimb bud h, heart; hl, hindlimb bud; paa, pharyngeal arch artery. Scale bar: 500 µm in (**A**), 100 µm in (**C**,**D**).

**Figure 2 ijms-23-07713-f002:**
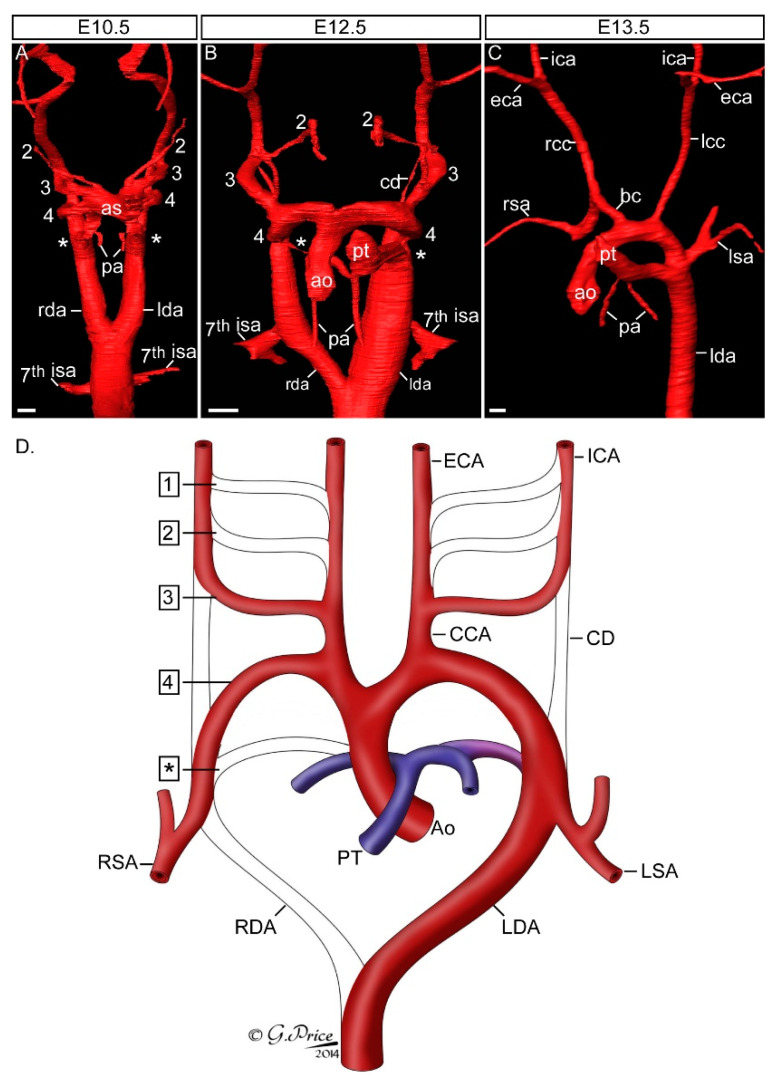
Development of the aortic arch arteries. Mouse embryos were processed for high resolution episcopic microscopy [9] and the acquired datasets used to make three-dimensional reconstructions with Amira software. (**A**) At the E10.5 stage the mouse embryo has symmetrical pairs of pharyngeal arch arteries. The first arch artery has remodelled and is no longer visible, the second is interrupted, and the third, fourth and ultimate (*) arch arteries are symmetrical, of equal size and connect the aortic sac (as) with the paired left and right dorsal aorta (lda, rda). The future subclavian arteries, the 7th intersegmental arteries (isa), emanate from the paired dorsal aorta near the point of bifurcation. (**B**) By E12.5 remodelling of the pharyngeal arch arteries is underway. The outflow tract has separated into the aorta (ao) and pulmonary trunk (pt), the ultimate artery (*) has thinned on the right and expanded on the left. The region of the dorsal aorta between the third and fourth arteries, the carotid duct (cd), is involuting. The right dorsal aorta caudal to the 7th intersegmental artery is regressing. (**C**) At the fetal stage, E13.5 in the mouse, the arch arteries have completed remodelling to produce the mature aortic arch arteries configuration. (**D**) Schematic showing the pharyngeal arch arteries (numbered, on the (**left**)) and the arteries they develop into (labelled, on the (**right**)). Abbreviations: ao, aorta; as, aortic sac; bc, brachiocephalic artery; cca, common carotid artery; cd, carotid duct; eca, external carotid artery; ica, internal carotid artery; isa, intersegmental artery; lcc, left common carotid; lda, left descending aorta; lsa, left subclavian artery; pa, pulmonary artery; pt, pulmonary trunk; rcc, right common carotid; rda, right descending aorta; rsa, right subclavian artery. Scale bar, 100 µm. Figure adapted from [7].

**Figure 3 ijms-23-07713-f003:**
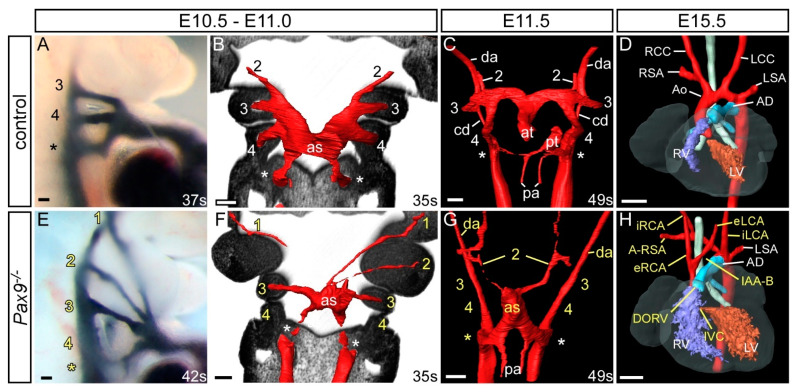
Cardiovascular developmental defects in *Pax9*-null mouse embryos. Control (**A**–**D**) and *Pax9*-null (*Pax9^−/−^*; **E**–**H**) mouse embryos were collected at different developmental stages and analysed for cardiovascular defects using intracardiac ink injections (**A**,**E**), high resolution episcopic microscopy (**B**,**C**,**F**,**G**) and magnetic resonance imaging (**D**,**H**). Control embryos at E10.5-E11.0 have the 3rd, 4th and ultimate (*) arch arteries patent to ink, are symmetrical in appearance and the 1st and 2nd have remodelled at this stage (**A**,**B**). In equivalently staged *Pax9*-null embryos the arch arteries are abnormal, with the 1st and 2nd persisting, the 3rd thin and the 4th absent (**E**,**F**). At E11.5 the caudal arch arteries of control embryos start to remodel, with the outflow tract septated and the right ultimate arch artery (*) regressing (**C**), whereas in *Pax9*-null embryos outflow tract septation is delayed, the 3rd arch arteries have collapsed and the 1st and/or 2nd persist (**G**). By E15.5 the pharyngeal arch arteries have remodelled into the adult aortic arch artery configuration (**D**). *Pax9*-null embryos show multiple defects (**H**) including aberrant right subclavian artery (A-RSA), interrupted aortic arch (IAA), abnormal right and left internal and external carotid arteries (i/eRCA, i/eLCA) and double outlet right ventricle (DORV) with interventricular communication (IVC). Somite numbers (s) are indicated. Abbreviations: AD, arterial duct; as, aortic sac; at, aortic trunk; cd, carotid duct; da, dorsal aorta; LCC, left common carotid artery; LSA, left subclavian artery; LV, left ventricle; pa, pulmonary artery; pt, pulmonary trunk; RCC, right common carotid artery; RSA, right subclavian artery; RV, right ventricle. Scale bar: 100 µm in (**A**–**C**) and (**E**–**G**), 500 µm in (**D**,**H**). Figure adapted from [88].

**Figure 4 ijms-23-07713-f004:**
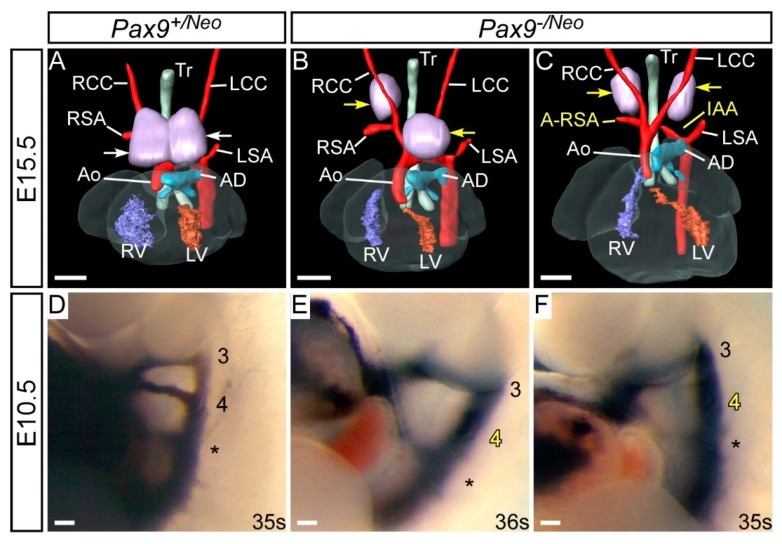
Developmental defects in *Pax9*-hypomorphic mouse embryos. *Pax9^+/Neo^* control (**A**,**D**) and *Pax9^−/Neo^* mutant (**B**,**C**,**E**,**F**) mouse embryos were collected and analysed for cardiovascular defects using magnetic resonance imaging at E15.5 (**A**–**C**) and intracardiac ink injections at E10.5 (**D**–**F**). (**A**) Control embryo with normal cardiovascular anatomy and thymus lobes (white arrows). *Pax9^−/Neo^* mutant embryos have normal arch arteries (**B**) or display fourth pharyngeal arch artery derived defects such as aberrant right subclavian artery and interruption of the aortic arch (**C**). The thymus lobes are frequently abnormal (yellow arrows; (**B**,**C**)). (**D**) The third (3), fourth (4) and ultimate (*) pharyngeal arch arteries are of equivalent size and patent to ink in control embryos. (**E**,**F**) In stage-matched *Pax9^−/Neo^* mutant embryos the fourth arch arteries are frequently absent. Somite numbers (s) are indicated. Abbreviations: AD, arterial duct; Ao, aorta; A-RSA, aberrant right subclavian artery; IAA, interrupted aortic arch; RCC, LCC, right/left common carotid artery; RSA, LSA, right/left subclavian artery; RV, LV, right/left ventricle; Tr, trachea. Scale bar: 500 µm in (**A**–**C**), 100 µm in (**D**–**F**).

**Figure 5 ijms-23-07713-f005:**
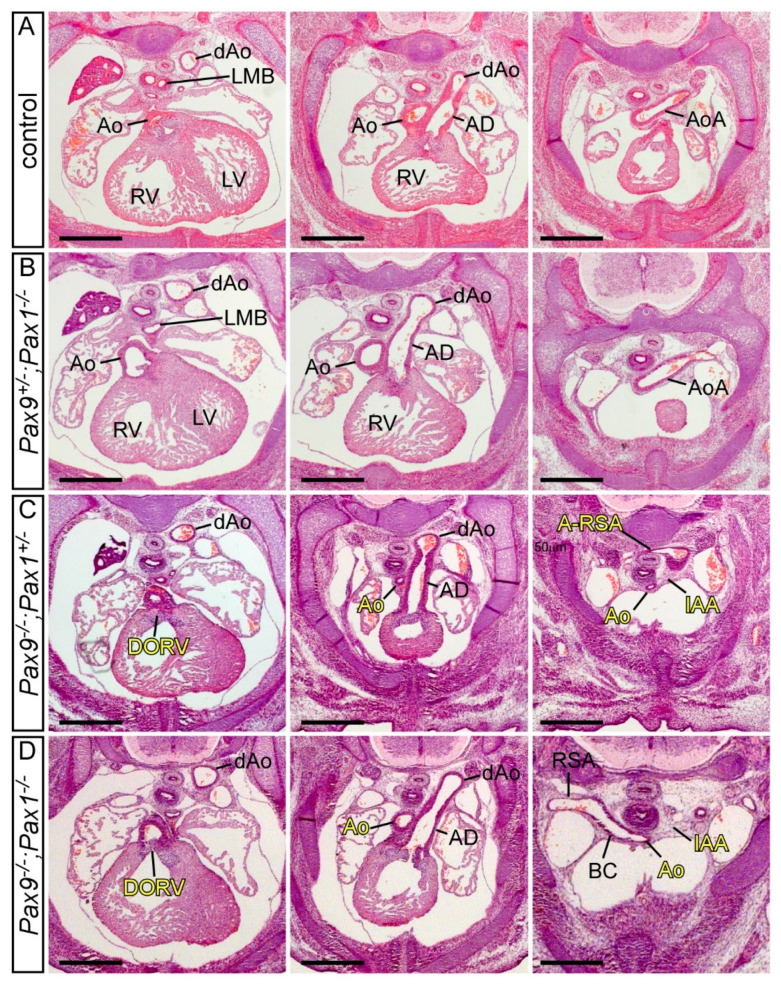
*Pax1* and *Pax9* do not genetically interact in cardiovascular development. Transverse haematoxylin and eosin-stained sections of E14.5 embryos. (**A**) Control embryo with normal outflow tract and arch artery morphology with the aorta arising from the left ventricle and the aortic arch crossing over the left main bronchus to join the dorsal aorta. (**B**) Embryo null for *Pax1* heterozygous for *Pax9* (*Pax9^+/−^; Pax1^−/−^*) with normal cardiovascular development comparable to the control. (**C**) Embryo heterozygous for *Pax1* and null for *Pax9* (*Pax9^−/−^;Pax1^+/−^*) displaying the typical cardiovascular defects seen in *Pax9*-null embryos. The aorta arises aberrantly from the right ventricle producing a double outlet right ventricle (DORV). The aorta is hypoplastic, with an interruption of the aortic arch (IAA) and an aberrant retro-oesophageal right subclavian artery (A-RSA). (**D**) Homozygous *Pax9;Pax1* null embryo (*Pax9^−/−^;Pax1^−/−^*) also with the typical Pax9-null cardiovascular defects, except with a normal right subclavian artery. Abbreviations: Ao, aorta; AoA, aortic arch; AD, arterial duct; BC, brachiocephalic artery; dAo, dorsal aorta; LMB, left main bronchus; LV, left ventricle; RSA, right subclavian artery; RV, right ventricle. Scale bar: 500 μm.

**Table 1 ijms-23-07713-t001:** Structure and expression of *Pax* genes discussed in this review.

Gene	Protein Domains	Class	Human	Mouse	Expression	Human Cardiac Defects
Paired	Oct	Homeo
*Pax1*	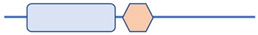	1	20p11	2	Pharyngeal endoderm, sclerotome	None
*Pax9*	14q12-13	12	PAD, IAA, BAV, VSD
*Pax3*	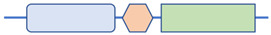	3	2q35	1	Neural crest cells, dermomyotome	ASD, VSD
*Pax7*	1p36	4	None

Abbreviations: ASD, atrial septal defect; BAV, bicuspid aortic valve; IAA, interrupted aortic arch; Homeo, paired-type homeodomain; Oct, octapeptide; PAD, patent arterial duct; VSD, ventricular septal defect.

**Table 2 ijms-23-07713-t002:** *Pax3* mutant alleles.

Allele	Mutation in *Pax3*	Cardiovascular Defect When Homozygous	Reference
*Sp*	Splice acceptor site in intron 3. Exon 4 not translated.	CAT, VSD	[40,45]
*Sp1H*	32bp deletion in exon 5	CAT, DORV, VSD, abnormal arch arteries *^a^*	[42]
*Sp2H*		CAT, DORV, VSD, arch arteries, congestive heart failure *^a^*	[43,44]
*Sp4H*	Deletion on Chromosome 1 between *Epha4* and *Cul3* (3.1 Mb)	ND, early embryonic death	[46]
*Spd*	*Pax3 ^G42R^*	DORV, VSD	[26]
*Spr*	Deletion on Chromosome 1 between *Vil1* and *Akp3* (12.6 Mb)	ND, early embryonic death	[47]
*Pax3* * ^Δ^ * * ^5^ *	Deletion of exon 5	CAT, VSD, absent left ultimate arch artery	[20]
*Pax3* * ^Δ^ * * ^5^ * *;AP-2aCre*	Deletion of exon 5 in NCC	DORV, VSD	
*Pax3Cre*	Replacement of exon 1 with Cre	CAT, DORV, VSD *^a^*	[48]
*Pax3^neo^*	Neomycin cassette inserted in intron 5 (hypomorphic allele)	No cardiovascular defects	[49]
*Pax3^neo^;Pax3* * ^Δ^ * * ^5^ * *;Pax7^+/−^*	*Pax3* hypomorph (10%) and *Pax7* heterozygous (50%)	CAT, DORV, VSD *^a^*	[50]
*Pax3^neo^;Pax3* * ^Δ^ * * ^5^ * *;Pax7^−/−^*	*Pax3* hypomorph (10%) and *Pax7*-null	CAT, VSD	
*Pax3^NCE^*	Deletion of NCC enhancer element	CAT, VSD	[51]
*Pax3-FKHR*	Pax3-Foxo1 fusion protein	CAT, DORV, VSD	[52]

*^a^* defects are not fully penetrant. Abbreviations: CAT, common arterial trunk; DORV, double outlet right ventricle; NCC, neural crest cell; ND, not determined; TV, tricuspid valve; VSD, ventricular septal defect.

## Data Availability

Not applicable.

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
