# Peer review of "PAX Genes in Cardiovascular Development"

_ijms, 2022, doi:10.3390/ijms23147713_

Round 1

Reviewer 1 Report

Rebecca et al have described the   PAX Genes in Cardiovascular Development with updated literature. This review is a nicely written manuscript with a detailed explanation of the PAX family genes involved in mouse heart development and phenotypic observations. However, the authors need to be addressed the following concerns before publication

Concerns

 1. Pax3 transcripts were disappears around day E17.0, need to explain clearly

2. In Table 2 authors need to add one column stating Human cardiac defects due to the Pax family of gene mutations.

Author Response

We thank the reviewer for their positive comments on our paper.

Concerns

  1. Pax3 transcripts were disappears around day E17.0, need to explain clearly

We have modified the sentence to clarify this:

Pax3 transcripts, however, are no longer expressed by E17.5 [24,25].

  1. In Table 2 authors need to add one column stating Human cardiac defects due to the Pax family of gene mutations.

We have now added a column to Table 1 to indicate the human cardiac defects associated with PAX gene mutation.

Reviewer 2 Report

The review manuscript by Steele et al. nicely describes the current state-of-the art knowledge on the role of Pax genes in cardiovascular development, particularly those exerted by Pax3 and Pax9. They provide a very comprehensive analyses of these genes in a manuscript that it is very well structured and easy to follow.

I do have only minor comments

Figure 3 legend seems to be misplaced and has become part of the main body text. Please modify accordingly.

The authors extensively describe some Pax9 plausible interacting transcription factors during cardiovascular development, but they fail to do the same with Pax3. It would be nice if the can add such information, if available, or if not, comment such shortcuts in the Pax3 section.

Author Response

We thank the reviewer for their positive comments on our paper.

Figure 3 legend seems to be misplaced and has become part of the main body text. Please modify accordingly.

This has been modified.

The authors extensively describe some Pax9 plausible interacting transcription factors during cardiovascular development, but they fail to do the same with Pax3. It would be nice if they can add such information, if available, or if not, comment such shortcuts in the Pax3 section.

We have now added new paragraphs relating to the interaction of Pax3 with Foxd3, Msx2, Sox10 and Tbx18.

Reviewer 3 Report

The review by Steele et al. describes the role that PAX genes play in cardiovascular development, with a focus on mouse models. Overall, I find this review very well written. The figures are very informative and appropriate to the topic. The role of PAX genes in outflow tract development, albeit known, is not always widely recognized so I think this review is timely to spread this information in the cardiovascular community. The topic is also well fitted to IJMS and its readers will find it of interest. My recommendation is for acceptance.

Author Response

We thank the reviewer for their positive comments on our paper.

No amendments were requested.